# Antimicrobial Diterpene Alkaloids from an *Agelas citrina* Sponge Collected in the Yucatán Peninsula

**DOI:** 10.3390/md20050298

**Published:** 2022-04-28

**Authors:** Dawrin Pech-Puch, Abel M. Forero, Juan Carlos Fuentes-Monteverde, Cristina Lasarte-Monterrubio, Marta Martinez-Guitian, Carlos González-Salas, Sergio Guillén-Hernández, Harold Villegas-Hernández, Alejandro Beceiro, Christian Griesinger, Jaime Rodríguez, Carlos Jiménez

**Affiliations:** 1Departamento de Química, Facultade de Ciencias e Centro de Investigacións Científicas Avanzadas (CICA), Universidade de A Coruña, 15071 A Coruña, Spain; dawrin.pech@correo.uady.mx (D.P.-P.); mateo.forerot@udc.es (A.M.F.); 2Departamento de Biología Marina, Universidad Autónoma de Yucatán, Km. 15.5, Carretera Mérida-Xmatkuil, A.P. 4-116 Itzimná, Merida C.P. 97100, Yucatán, Mexico; carlos.gonzalez@correo.uady.mx (C.G.-S.); ghernand@correo.uady.mx (S.G.-H.); harold.villegas@correo.uady.mx (H.V.-H.); 3Department of NMR Based Structural Biology, Max Planck Institute (MPI) for Multidisciplinary Sciences, Am Fassberg 11, 37077 Göttingen, Germany; jufu@mpinat.mpg.de (J.C.F.-M.); cigr@mpinat.mpg.de (C.G.); 4Microbiology Department of the University Hospital A Coruña (CHUAC), Institute of Biomedical Research of A Coruña (INIBIC), Centro de Investigación Biomédica en Red (CIBER) Infec., 15006 A Coruña, Spain; crlasarm@gmail.com (C.L.-M.); m.martinez.guitian@gmail.com (M.M.-G.); alejandro.beceiro.casas@sergas.es (A.B.)

**Keywords:** diterpene alkaloids, agelasines, agelasidines, marine sponge, *Agelas citrina*, antibacterial

## Abstract

Three new diterpene alkaloids, (+)-8-epiagelasine T (**1**), (+)-10-epiagelasine B (**2**), and (+)-12-hydroxyagelasidine C (**3**), along with three known compounds, (+)-*ent*-agelasine F (**4**), (+)-agelasine B (**5**), and (+)-agelasidine C (**6**), were isolated from the sponge *Agelas citrina*, collected on the coasts of the Yucatán Peninsula (Mexico). Their chemical structures were elucidated by 1D and 2D NMR spectroscopy, HRESIMS techniques, and a comparison with literature data. Although the synthesis of (+)-*ent*-agelasine F (**4**) has been previously reported, this is the first time that it was isolated as a natural product. The evaluation of the antimicrobial activity against the Gram-positive pathogens *Staphylococcus aureus*, *Streptococcus pneumoniae*, *Enterococcus faecalis* showed that all of them were active, with (+)-10-epiagelasine B (**2**) being the most active compound with an MIC in the range of 1–8 µg/mL. On the other hand, the Gram-negative pathogenes *Acinetobacter baumannii*, *Pseudomonas aeruginosa*, and *Klebsiella pneumoniae* were also evaluated, and only (+)-agelasine B (**5**) showed a moderate antibacterial activity with a MIC value of 16 μg/mL.

## 1. Introduction

One of the most common sponges in tropical and subtropical areas around the world are marine sponges of the genus *Agelas* (class Demospongiae, order Agelasida, family Agelasidae) [1]. Covering the period from 1971 to November 2021, 355 compounds have been isolated from *Agelas* sponges with interesting biological activities [2]. Many of these metabolites are of mixed biogenetic origins, as illustrated by alkaloids (especially pyrrole alkaloids and terpenoid alkaloids) [3,4], glycosphingolipids, sterols, and carotenoids that display diverse biological activities and unique structural features [5,6,7]. 

In particular, the nitrogenated diterpenes that include hypotaurocyamines (agelasidines) and *N*^9^-adeninium alkaloids (agelasines) represent an important class of compounds isolated from this genus and they show interesting biological activities, including antibacterial, cytotoxic, antifouling, antifungal, and inhibitory effects on Na^+^/K^+^-ATPase [8].

As part of our ongoing efforts to find new bioactive natural compounds from marine organisms collected off the coast of the Yucatán Peninsula (Mexico) [9,10,11,12], specimens of the sponge *A. citrina* were investigated due to the antibacterial activity displayed by their organic extracts. Former studies of the sponge *A. citrina* reported the presence of diterpene alkaloids [13] and pyrrole—imidazole alkaloids [14]. Our preliminary chemical investigation of this sponge led to the isolation of the major known component (−)-agelasine B, which showed significant antibacterial activity against two Gram-positive bacteria *S. aureus* strains [10]. Further bioactivity-guided fractionation of the CH_2_Cl_2_ and aqueous methanolic fractions led to the isolation of three new compounds (**1**, **2**, and **3**), as well as three known compounds (**4**, **5**, and **6**). All isolated compounds (**1**–**6**) were evaluated for their antibacterial activity. Herein, we describe the isolation, structure elucidation, and antibacterial activity of **1**–**6**.

## 2. Results and Discussion

### 2.1. Isolation and Identification of Agelasines

Specimens of the sponge *A. citrina* collected on Cozumel Island of the Yucatán Penisula (Mexico) were extracted several times with CH_3_OH/CH_2_Cl_2_ to produce an organic extract, which was partitioned following our standard partitioning procedure [15] into several fractions of differing polarities. The CH_2_Cl_2_ and aqueous methanolic fractions were submitted to Solid Phase Extraction (SPE) with RP-18 cartridges and finally to RP-HPLC to produce pure compounds **1**–**6** (Figure 1).

Compound **1** was obtained as a yellow oil [α]^25^_D_ = +12 (c 0.36, MeOH). The molecular formula of **1** was established based on the [M]^+^ peak at *m*/*z* 440.3375, observed in its (+)-HRESIMS (calculated for C_26_H_42_N_5_O, *m*/*z* 440.3384, Δ = 2.04 ppm).

The ^1^H and ^13^C NMR spectral data, along with an HSQC spectrum of **1** in DMSO-*d*_6_ (Table 1), were indicative of an agelasine-type diterpene alkaloid compound. The adenine moiety was identified by the characteristic signals of two methines at *δ*_H_*/δ*_C_ 8.46 (1H, s, H-2′)/155.4 (C-2′) and 9.51 (1H, s, H-8′)/140.8 (C-8′); a *N*-methyl group at *δ*_H_*/δ*_C_ 3.88 (3H, s)/31.1; and three non-protonated sp^2^ carbons at *δ*_C_ 152.4 (C-6′), 149.0 (C-4′), and 109.2 (C-5′). The diterpene portion of **1** was assigned from the ^1^H and ^13^C NMR signals of the bicyclic ring corresponding to four diasterotopic methylenes observed at *δ*_H_*/δ*_C_ 1.58 (1H, m, H-1a)-0.89 (1H, m, H-1b)/39.2 (C-1), 1.53 (1H, m, H-2a)-1.19 (1H, m, H-2b)/19.8 (C-2), 1.33 (1H, m, H-3a)-1.10 (1H, m, H-3b)/41.3 (C-3), and 1.70 (1H, dt, *J* = 11.8, 3.0 Hz, H-7a)-1.30 (1H, m, C-7b)/43.7 (C-7); one homotopic methylene at *δ*_H_/*δ*_C_ 1.34 (2H, m, H-6)/17.7 C-6; two methines at *δ*_H_/*δ*_C_ 0.81 (1H, m, H-5)/55.3 (C-5) and 0.98 (1H, m, H-9)/59.9 (C-9); four methyl groups at *δ*_H_/*δ*_C_ 1.00 (3H, s, H_3_-17)/23.6 (C-17), 0.84 (3H, s, H_3_-18)/33.0 (C-18), 0.75 (3H, s, H_3_-19)/21.1 (C-19), and 0.74 (3H, s, H_3_-20)/15.0 (C-20); two quaternary carbons at *δ*_C_ 32.9 (C-4) and 38.5 (C-10); and one non-protonated carbon linked to oxygen at *δ*_C_ 72.1 (C-8). Moreover, the ^1^H and ^13^C NMR signals corresponding to three methylenes at *δ*_H_/*δ*_C_ 1.55 (1H, m, H-11a)-1.27 (1H, m, H-11b)/22.7 (C-11), 2.22 (1H, td, *J* = 12.2, 6.1 Hz, H-12a)-2.09 (1H, td, *J* = 12.2, 4.9 Hz, H-12b)/42.2 (C-12), and 5.13 (2H, d, *J* = 7.3 Hz, H-15)/46.6 (C-15); a vinyl methyl at *δ*_H_/*δ*_C_ 1.79 (3H, s, H_3_-16)/16.5 (C-16); an olefinic methine at *δ*_H_/*δ*_C_ 5.44 (1H, t, *J* = 7.0 Hz, H-14)/114.6 (CH); and a quaternary sp^2^ carbon at *δ*_C_ 147.0 (C-13) confirmed the presence of the 3-methylpentenyl chain.

The analysis of COSY and HMBC spectra was able to confirm the planar structure of **1** (Figure 2). The 1D NMR spectral data of **1** were similar to those for agelasine T, a previously reported agelasine isolated from an unidentified *Agelas* collected in Okinawa [16]. However, the difference of the carbon chemical shift of the C-17 methyl group at *δ*_C_ 23.6 in **1** instead of the reported value of that group at *δ*_C_ 30.7 in agelasine T, suggested that they differed in the spatial disposition of the C-17 methyl group, and consequently in the stereochemistry at C-8 position.

The relative configuration was assigned by 2D and selective 1D-NOESY experiments of **1**. The *E*-geometry of the Δ^13^ double bond was established by the observed NOESY correlation between the signals assigned to the methylene CH_2_-15 at *δ*_H_ 5.13 and vinyl methyl CH_3_-16 at *δ*_H_ 1.79. On the other hand, the irradiation of methyl protons (Appendix A) at the position CH_3_-17 (*δ*_H_ 1.00) showed a 2.0% NOE response to the methyl protons at the position CH_3_-20 (*δ*_H_ 0.74); the irradiation of methyl protons (Appendix A) at the position CH_3_-18 (*δ*_H_ 0.84) also displayed NOE to the methyl protons at the position CH_3_-20 (*δ*_H_ 0.74) (3.0%) (see Figure 3A). Thus, these NOE correlations suggest that these three methyl groups are located on the same face of the molecule, assigned as the β face. Furthermore, the carbon chemical shift of CH_3_-17 (*δ*_C_ 23.6) resonates similarly to that of a synthetic terpene having a bicyclic ring bearing the same spatial disposition of the hydroxy group (*δ*_C_ 24.2) [17].

This analysis makes it possible to propose the structure of compound **1** as (+)-8-epiagelasine T (Figure 1).

Compound **2** was obtained as a yellow oil [α]^25^_D_ = +16 (c 0.13, MeOH) and its molecular formula was established on the basis of the [M]^+^ peak at *m*/*z* 422.3280, observed in its HRESIMS spectrum (calculated for C_26_H_40_N_5_, *m/z* 422.3279, Δ = 0.23 ppm).

The similarity of ^1^H and ^13^C NMR spectral in CDCl_3_ of **2** to those of **1** suggested that **2** was also an agelasine-type alkaloid with an *N*^9^-methyladenine unit. 

The adenine moiety was identified by the characteristic signals of two methines at *δ*_H_*/δ*_C_ 8.54 (1H, s, H-2′)/155.8 (C-2′) and 9.87 (1H, s, H-8′)/142.2 (C-8′), an *N*-methyl group at *δ*_H_*/δ*_C_ 4.06 (3H, s)/31.9, and three non-protonated sp^2^ carbons at *δ*_C_ 151.8 (C-6′), 149.7 (C-4′) and 110.0 (C-5′). The bicyclic ring for the diterpene portion of **2** was assigned from the ^1^H and ^13^C NMR signals corresponding to four diasterotopic methylenes observed at *δ*_H_*/δ*_C_ 1.64 (1H, m, H-1a)-1.33 (1H, m, H-1b)/19.7 (C-1), 1.98 (1H, m, H-2a)-1.91 (1H, m, H-2b)/22.9 (C-2), 1.80 (1H, m, H-6a)-1.71 (1H, m, H-6b)/33.7 (C-6), and 1.64 (1H, m, H-7a)-1.42 (1H, m, H-7b)/29.2 (C-7); an olefinic methine at *δ*_H_*/δ*_C_ 5.30 (1H, brs, H-3)/120.7 (C-3); two aliphatic methines at *δ*_H_*/δ*_C_ 1.31 (1H, m, H-8)/33.5 (C-8) and 2.28 (1H, brd, *J* = 12.9 Hz, H-10)/42.0 (C-10); a vinyl methyl at *δ*_H_*/δ*_C_ 1.62 (3H, s, H_3_-18)/21.7 (C-18); three methyl groups at *δ*_H_*/δ*_C_ 1.07 (3H, d, *J* = 7.4 Hz, H_3_-17)/18.7 (C-17), 0.86 (3H, s, H_3_-19)/17.2 (C-19), and 0.94 (3H, s, H_3_-20)/15.6 (C-20); two quaternary carbons at *δ*_C_ 40.5 (C-5) and 39.1 (C-9); and a quaternary sp^2^ carbon at *δ*_C_ 136.2 (C-4). Additionally, the ^1^H and ^13^C NMR signals corresponding to three methylenes at *δ*_H_/*δ*_C_ 1.35 (1H, m, H-11a)-1.25 (1H, m, H-11b)/30.0 (C-11), 2.03 (1H, dt, *J* = 11.8, 5.6 Hz, H-12a)-1.97 (1H, m, H-12b)/35.1 (C-12), and 5.25 (2H, d, *J* = 6.5 Hz, H-15)/48.6 (C-15); a vinyl methyl at *δ*_H_*/δ*_C_ 1.86 (3H, s, H_3_-16)/17.2 (C-16); an olefinic methine at *δ*_H_*/δ*_C_ 5.45 (1H, t, *J* = 6.4 Hz, H-14)/115.0 (C-14); and a quaternary sp^2^ carbon at *δ*_C_ 149.2 (C-13) confirmed the presence of the 3-methylpentenyl chain.

The constitution of **2** was established from the analysis of the 2D NMR spectral data (Figure 2). Two spin systems corresponding to the bicyclic system were deduced from the COSY experiment of **2**: the first spin system was assigned to the H-1, H-2, H-3, and H-10 protons, and the second spin system to the H-6, H-7, H-8, and H_3_-17 protons (Figure 2). Key HMBC correlations from H_3_-19 (*δ*_H_ 0.86) to C-4 (*δ*_C_ 136.2), C-5 (*δ*_C_ 40.5)/C-6 (*δ*_C_ 33.7) and C-10 (*δ*_C_ 42.0); from H_3_-17 (*δ*_H_ 1.07) to C-8 (*δ*_C_ 33.5), C-9 (*δ*_C_ 39.1) and C-7 (*δ*_C_ 29.2); and from H_3_-20 (*δ*_H_ 0.94) to C-8 (*δ*_C_ 33.5), C-9 (*δ*_C_ 39.1) and C-10 (*δ*_C_ 42.0) allowed us to connect the two spin systems. Finally, the HMBC correlation from H_3_-20 (*δ*_H_ 0.94) to C-11 (*δ*_C_ 30.0) showed the link between the bicyclic ring and the 3-methylpentenyl chain (Figure 2). All these data indicate the presence of a clerodane skeleton in **2** as in agelasine B (**5**), also isolated from this sponge. 

The relative configuration of **2** was deduced from NOESY data and confirmed by a IPAP-HSQMBC experiment. The *E*-configuration of the Δ^13^ double bond of **2**, as in agelasine B (**5**), was deduced from NOE correlations observed in the NOESY experiment between the signals assigned to the methylene CH_2_-15 at *δ*_H_ 5.25 and vinyl methyl CH_3_-16 at *δ*_H_ 1.86, and between the methylene CH_2_-12 at *δ*_H_ 2.03 and ofefinic proton CH-14 at *δ*_H_ 5.45. 

On the other hand, the strong NOE correlations from H_3_-17 (*δ*_H_ 1.07) to H_3_-19 (*δ*_H_ 0.86) and H_3_-20 (*δ*_H_ 0.94) in the NOESY experiment of **2** indicate that these methyl groups (H_3_-17, H_3_-19, and H_3_-20) are oriented on the same face of the molecule, assigned as β-face. 

The drastic change in the chemical shift of H-10 in **2** at *δ*_H_ 2.28 (1H, brd, *J* = 12.9 Hz) in relation to that in agelasine B (**5**) at *δ*_H_ 1.29 (1H, m) (Table 1), and considering that agelasine B (**5**) has the same spatial disposition for the three methyl groups H_3_-17, H_3_-19, and H_3_-20, suggested they must differ in the remaining stereogenic center at C-10. Similar chemical shifts and the same HMBC and NOESY correlations were observed in **2** when the NMR spectra were run in C_6_D_6_ (Appendix A). Therefore, these results suggest a *cis*-clerodane disposition for compound **2**, instead of the *trans*-clerodane disposition for agelasine B (**5**). 

Curiously, when the NMR spectra of **2** was repeated this time in CDCl_3_, an important displacement of the carbon and proton chemical shifts was observed in the NMR spectra after 24 h, probably due to the acidic character of the deuterated solvent. Thus, the carbon and proton chemical were assigned again by 2D-NMR experiments (see Appendix A and Appendix A). The low value of the ^3^*J*_CH_ between H-10 and C-19 of 3.42 Hz measured in an IPAP-HSQMBC experiment of this sample (Appendix A) agrees with the proposed *cis*-clerodane disposition for **2**. Unfortunately, the ^3^*J*_CH_ between H-10 and C-19 in agelasine B (**5**) could not be determined due to the insufficient amount. In order to obtain a direct proof for the *cis*- vs. *trans*-ring connections in decaline-like systems using the IPAP-HSQMBC experiment, we measured the corresponding values in the available synthetic model compounds **7** and **8** (Figure 4) [18]. In this way, the corresponding value of the *cis*-decalin-like system in **7** (^3^*J*_C7H5_ = 2.9 Hz) resulted to be similar to that of **2** (Appendix A). As expected, the corresponding value of the *trans*-decalin-like system (^3^*J*_C7H5_ = 8.42 Hz) was much larger (Appendix A).

On the other hand, the selective 1D-NOESY correlations from H_3_-17 (*δ*_H_ 1.10) to H_3_-20 (*δ*_H_ 0.87) (0.30%), H_3_-18 (*δ*_H_ 1.62) (0.36%) and H-10 (*δ*_H_ 1.25) (0.23%) (Appendix A); from H_3_-19 (*δ*_H_ 1.15) to H_3_-20 (*δ*_H_ 0.87) (0.5%) (Appendix A); and from H_3_-20 (*δ*_H_ 0.87) to H_3_-17 (*δ*_H_ 1.10) (0.25%), H_3_-19 (*δ*_H_ 1.15) (0.32%), and H-10 (*δ*_H_ 1.25) (0.30% (Appendix A), indicating that H-10, H_3_-17, H_3_-19, and H_3_-20 protons are in the same face of the molecule, confirmed the *cis*-clerodane disposition for compound **2** (Figure 3B). Thus, we named this new agelasine as 10-epiagelasine B (**2**).

The molecular formula of **3**, also obtained as a yellow oil [α]^25^_D_ = + 2 (c 0.2, MeOH), was established from its (+)-HRESIMS, which displayed a [M + H]^+^ adduct at *m/z* 440.2950 (calculated for C_23_H_42_N_3_SO_3_, *m/z* 440.2942, Δ = 2.04 ppm) and from its ^13^C NMR spectrum.

The ^1^H-NMR and ^13^C NMR spectral data of **3** in CDCl_3_ (Table 2) were indicative of an agelasidine-type diterpene alkaloid compound. The hypotauro-cyamine moiety (-SO_2_-CH_2_CH_2_NHC(NH)NH_3_^+^) was identified by the characteristic guanidine carbon signal at *δ*_C_ 157.5 (C-3′) and two methylenes at *δ*_H_*/δ*_C_ 3.28 (2H, brs, H-1′)/0.3 (C-1′) and 3.70 (2H, brs, H-2′)/34.5 (C-2′). The diterpene portion of **3** was assigned by the ^1^H and ^13^C NMR signals of the monocyclic ring corresponding to two homotopic methylenes observed at *δ*_H_*/δ*_C_ 1.44 (2H, m, H-1)/27.2 (C-1) and 1.95 (2H, m, H-2)/25.7 (C-2); an olefinic methine at *δ*_H_*/δ*_C_ 5.40 (1H, brs, H-3)/124.4 (C-3); an aliphatic methine at *δ*_H_*/δ*_C_ 1.71 (1H, m, H-6)/33.3 (C-6); a vinyl methyl at *δ*_H_*/δ*_C_ 1.60 (3H, brs, H_3_-18)/19.3 (C-18); two methyl groups at *δ*_H_*/δ*_C_ 0.84 (3H, s, H-19)/16.0 (C-19) and 0.85 (3H, d, *J* = 6.7 Hz, H-20)/21.2 (C-20); a quaternary carbon at *δ*_C_ 40.5 (C-5); and a quaternary sp^2^ carbon at *δ*_C_ 139.7 (C-4). Furthermore, the ^1^H and ^13^C NMR signals corresponding to four methylenes at *δ*_H_/*δ*_C_ 1.43 (2H, m, H-7)/35.3 (C-7), 1.95 (1H, m, H-8a)-1.48 (1H, m, H-8b)/34.6 (C-8), 2.27 (2H, t, *J* = 6.5 Hz, H-11)/34.0 (C-11), and 3.84 (1H, dd, *J* = 14.5, 8.0 Hz, H-15a)-3.80 (1H, dd, *J* = 14.5, 8.0 Hz, H-15b)/53.8 (C-15); two vinyl methyls at *δ*_H_/*δ*_C_ 1.71 (3H, brs, H-16)/13.1 (C-16) and 1.61 (1H, brs, H-17)/16.6 (C-17); two olefinic methines at *δ*_H_/*δ*_C_ 5.06 (1H, t, *J* = 6.5 Hz, H-10)/118.6 (C-10) and 5.52 (1H, t, *J* = 8.0 Hz, H-14)/110.0 (C-14); and two quaternary sp^2^ carbons at *δ*_C_ 140.1 (C-9) and 149.2 (C-13) confirmed the presence of the 3,7-dimethyl-nona-3,7-dienyl chain.

The 1D NMR spectral data of **3** were similar to those of (+)-agelasidine C, isolated from *Agelas nakamurai* collected in Okinawa [19]. The presence of an additional hydroxyl group in **3** in relation to (+)-agelasidine C was suggested by the difference of an oxygen (16 Da) between their molecular formula. This was confirmed by the presence of an additional oxymethine at *δ*_H_/*δ*_C_ 4.09 (1H, t, *J* = 6.5 Hz, H-12)/76.3 (C-12) in the ^1^H and ^13^C NMR spectra of **3,** instead of the corresponding methylene group present in (+)-agelasidine C. The HMBC correlations between the vinyl methyl protons at *δ*_H_ 1.71 (3H, s, H_3_-16) and the oxymethine carbon at *δ*_C_ 76.3 (C-12), the quaternary sp^2^ carbon at *δ*_C_ 149.2 (C-13) and the olefinic methine carbon at *δ*_C_ 110.0 (C-14), located the hydroxyl group at C-12 position (Figure 2). The analysis of COSY and HMBC spectra was able to confirm the constitution of **3** (Figure 2). These data indicate that **3** is a new agelasidine, which was named 12-hydroxyagelasidine C.

The relative configuration of the monocyclic ring system in **3** was confirmed by a comparison of its ^13^C NMR spectral data with that of (+)-agelasidine C [19] and other agelasidines [20] having the same relative configuration. The *E*-configuration of the Δ^9^ and Δ^13^ double bonds was established from the NOESY correlation between the signals assigned to the methylene CH_2_-11 at *δ*_H_ 2.27 and the vinyl methyl CH_3_-17 at *δ*_H_ 1.61, and between the methylene CH_2_-15 at *δ*_H_ 3.84–3.80 and the vinyl methyl CH_3_-16 at *δ*_H_ 1.71, respectively. Absolute configuration at C-12 in **3** was determined by the Mosher’s method using the MTPA esters [21]. The 12-hydroxyagelasidine C (**3**) was treated with *R*-(−)- and *S*-(+)-α-methoxy-α-(trifluoromethyl) phenyl acetic acid (MTPA-OH) to afford the *S*- and *R*-MTPA esters. The analysis of the ^1^H NMR and ^1^H−^1^H COSY led to the assignment of both esters’ chemical shifts in proximity of C-12. The Δ*δ*^SR^ values between *R*- and *S*-MTPA esters of **3** at 12-OH were negative for H-11/10/17 and positive for H-14/15/18 (Figure 5), which suggested the absolute configuration of C-12 as *R*.

Compounds **4**–**6** (Figure 1) were identified as the known (+)-*ent*-agelasine F (**4**) [22], (+)-agelasine B (**5**) [8] and (+)-agelasidine C (**6**) [19], by comparing their NMR data with that reported in the literature. Although (+)-*ent*-agelasine F (**4**) have been previously synthetized, this is the first time that is reported as a natural product.

### 2.2. Antibacterial Activity of Agelasines

The antibacterial evaluation of **1**–**6** against the Gram-negative pathogens *A. baumannii* ATCC 17978, *K. pneumoniae* ATCC 700603, and *P. aeruginosa* ATCC 27823 displayed that only (+)-agelasine B (**5**) showed a moderate antibacterial activity with an MIC value of 16 μg/mL. In contrast, all of them exhibited antibacterial activity against Gram-positive pathogens (Table 3). These results are in accordance with previous reports, where (+)-*ent*-agelasine F (**4**), (+)-agelasine B (**5**) [8], and (+)-agelasidine C (**6**) [19] showed antimicrobial activity against Gram-positive bacteria.

Interestingly, the higher activity shown by (+)-10-epiagelasine B (**2**), having a *cis*-clerodane, than its isomer (+)-agelasine B (**5**), bearing a *trans*-clerodane, in Gram-positive pathogens, was indicative of the impact of decalin stereochemistry in the detected activity.

## 3. Materials and Methods

### 3.1. General Experimental Chemical Procedures

Optical rotations were measured on a JASCO DIP-1000 polarimeter, with an Na (589 nm) lamp and filter. ^1^H, ^13^C, and 2D NMR spectra were recorded on a Bruker spectrometer of 950 MHz, equipped with a 5 mm Cryo Probe (NEO console); a Bruker spectrometer (800 MHz for ^1^H and 200 MHz for ^13^C) equipped with a 5 mm cryo probe and an NEO console; a Bruker (800 MHz for ^1^H and 200 MHz for ^13^C) spectrometer equipped with a 3 mm cryo probe and a NEO console; a Bruker spectrometer (700 MHz for ^1^H and 175 MHz for ^13^C) equipped with a 5 mm cryo probe and an Avance III console; and a Bruker Avance 500 spectrometer (500 MHz for ^1^H and 125 MHz for ^13^C) equipped with a 5 mm cryo probe, using DMSO-*d*_6_ and CDCl_3_ as solvents. Chemical shifts are reported in *δ* scale relative to DMSO-*d*_6_ (*δ* 2.50 ppm for ^1^H NMR, *δ* 39.51 ppm for ^13^C NMR) and CDCl_3_ (*δ* 7.26 ppm for ^1^H NMR, *δ* 77.0 ppm for ^13^C NMR).

HRESIMS experiments were performed on the Applied Biosystems QSTAR Elite system or a Thermo MAT95XP spectrometer. HPLC separations were performed on the Agilent 1100 liquid chromatography system equipped with a solvent degasser, quaternary pump, and diode array detector (Agilent Technologies, Waldbronn, Germany) using a semipreparative reversed phase column Luna C18 (5 μ, 100 Å, 250 × 10 mm).

### 3.2. Animal Material

The sponge *A. citrina* was collected by hand and traditional SCUBA-diving off Cozumel Island, Quintana Roo (20°41′00.00″ N/87°01′32.66″ W) at depths ranging from 10 to 15 m in March 2017, and frozen immediately after collection. A voucher specimen CZE56 was deposited in the Phylum Porifera Gerardo Green National Collection of the Institute of Marine Sciences and Limnology (ICMyL) at the National Autonomous University of Mexico (UNAM), Mexico City.

### 3.3. Extraction and Isolation

Sliced bodies of *A. citrina* (wet weight, 729.6 g; dry weight, 375.3 g) were exhaustively extracted with CH_3_OH-CH_2_Cl_2_ (1:1, 3 × 1.5 L) at 25 °C for 24 h each maceration. The combined extracts were concentrated under reduced pressure to produce 6.1 g of a crude residue. A total of 6.0 g was first partitioned between CH_2_Cl_2_/H_2_O (1:1 *v*/*v*) to produce aqueous and organic phases. The organic phase was concentrated under reduced pressure and partitioned between 10% aqueous CH_3_OH (400 mL) and hexane (2 × 400 mL). The hexane portion produced, after removing the solvent under reduced pressure, 672.2 mg of the hexane fraction (FH). The H_2_O content (% *v*/*v*) of the methanolic fraction was adjusted to 50% aqueous CH_3_OH, and the mixture was extracted with CH_2_Cl_2_ (100 mL) to afford, after removing the solvent under reduced pressure, 3.6 g of the CH_2_Cl_2_ fraction (FD) and 755.8 mg of the remaining aqueous methanolic fraction (FM) (Appendix A).

The CH_2_Cl_2_ fraction (3.6 g) was subjected to a Solid Phase Extraction (SPE) with RP-18 (Merck KGaA) using a discontinuous gradient from H_2_O to CH_3_OH and then CH_2_Cl_2_. 

The fraction eluted with H_2_O/CH_3_OH (2:1, 181.3 mg) was separated by RP-HPLC eluting, with a mobile phase consisting of 30 min of a gradient from 30% to 100% of CH_3_OH in H_2_O (*v*/*v*, each containing 0.04% trifluoroacetic acid) followed by a 10 min isocratic at 100% of CH_3_OH at a flow rate of 2.0 mL/min, afforded (+)-8-epilagelasine T (**1**) (30.0 mg; R_t_ = 34.9 min) and (+)-10-epiagelasine B (**2**) (45.0 mg; R_t_ = 37.0 min).

The separation of the fraction eluted with H_2_O/CH_3_OH (1:2, 391.1 mg) was performed by RP-HPLC with a mobile phase consisting of 30 min of a gradient from 40% to 100% of CH_3_OH in H_2_O (*v*/*v*, each containing 0.04% trifluoroacetic acid), followed by a 10 min isocratic at 100% of CH_3_OH at a flow rate of 2.0 mL/min, yielded (+)-*ent*-agelasine F (**4**) (21.3 mg; R_t_ = 30.1 min) and (+)-agelasine B (**5**) (1.5 mg; R_t_ = 32.5 min).

The aqueous methanolic fraction (755.8 mg) was submitted to a Solid Phase Extraction (SPE) with RP-18 (Merck KGaA) using a discontinuous gradient from H_2_O to CH_3_OH and then CH_2_Cl_2_. The separation of the fraction eluted with H_2_O/CH_3_OH (1:1, 181.5 mg) by RP-HPLC using a mobile phase consisting of 30 min of a gradient from 35% to 100% of CH_3_OH in H_2_O (*v*/*v*, each containing 0.04% trifluoroacetic acid), followed by a 10 min isocratic at 100% of CH_3_OH at a flow rate of 2.0 mL/min, afforded (+)-12-hidroxy agelasidine C (**3**) (7.0 mg; R_t_ = 29.8 min) and (+)-agelasidine C (**6**) (27.0 mg; R_t_ = 34.0 min).

### 3.4. Structural Characterization

(+)-8-epiagelasine T (**1**). [α]^25^_D_ +12 (c 0.2, MeOH); ^1^H and ^13^C NMR see Table 1; (+)-HRESIMS *m*/*z* 440.3375 [M]^+^ (calcd. for C_26_H_42_N_5_O, 440.3384).

(+)-10-epiagelasine B (**2**). [α]^25^_D_ +16 (c 0.2, MeOH); ^1^H and ^13^C NMR see Table 1; (+)-HRESIMS *m*/*z* 422.3280 [M]^+^ (calcd. for C_26_H_40_N_5_, 422.3279).

(+)-12-hidroxyagelasidine C (**3).** [α]^25^_D_ +2 (c 0.2, MeOH); ^1^H and ^13^C NMR see Table 2; (+)-HRESIMS *m*/*z* 440.2950 [M + H]^+^ (calcd. for C_23_H_42_N_3_SO_3_, 440.2942).

(+)-*ent*-agelasine F (**4**). [α]^25^_D_ +1 (c 0.2, MeOH); ^1^H and ^13^C NMR see SI; (+)-HRESIMS *m*/*z* 422.3280 [M + H]^+^ (calcd. for C_26_H_40_N_5_, 422.3279).

(+)-agelasine B (**5**). [α]^25^_D_ +13 (c 0.2, MeOH); ^1^H and ^13^C NMR see SI; (+)-HRESIMS *m*/*z* 422.3281 [M + H]^+^ (calcd. for C_26_H_40_N_5_, 422.3284).

(+)-agelasidine C (**6**). [α]^25^_D_ +9 (c 0.2, MeOH); ^1^H and ^13^C NMR see SI; (+)-HRESIMS *m*/*z* 424.2991 [M + H]^+^ (calcd. for C_23_H_42_N_3_SO_2_, 424.2993).

### 3.5. Preparation of the MTPA Esters

The (*R*)- and (*S*)-MTPA esters were obtained by treatment of **3** (1.5 mg, for each acid) with (*R*)- and (*S*)-MTPA (1.60 mg), DCC (1.40 mg), and DMAP (1.0 mg), in dry CH_2_Cl_2_ (1.0 mL). The reactions were stirred at room temperature for 24 h and the resulting products were purified with SPE cartridges (Hypersep C18 200 mg, MeOH/H_2_O 50/50, *v*/*v*) to afford the *S*-(−)- and *R*-(+)-MTPA esters **3a** and **3b**. The products were monitored by recording ^1^H NMR spectra at 500 MHz:

^1^H NMR data of **3a** (500 MHz in CDCl_3_): *δ*_H_ 5.04 (t, 6.5, H-10), 2.22 (m, H-11), 4.94 (t, 6.5, H-12), 5.48 (t, 8.0, H-14), 3.73 (m, H-15a), 3.77 (m, H-15b), 1.58 (brs, CH3-17), 1.69 (brs, CH_3_-18).

^1^H NMR data of **3b** (500 MHz in CDCl_3_): *δ*_H_ 5.08 (t, 6.5, H-10), 2.30 (m, H-11), 4.77 (t, 6.5, H-12), 5.47 (t, 8.0, H-14), 3.69 (m, H-15a), 3.65 (m, H-15b), 1.60 (brs, CH3-17), 1.67 (brs, CH_3_-18).

### 3.6. Antibacterial Activity Assays

#### 3.6.1. Bacterial Strains and Culture Preparation

The bacterial strains used to study the antibacterial activity of the isolated compounds were the Gram-negative pathogens *Acinetobacter baumannii* (ATCC 17978), *Pseudomonas aeruginosa* (ATCC 27823), and *Klebsiella pneumoniae* (ATCC 700603), and the Gram-positive pathogens, including type-strains and clinical strains isolated from infections in our hospital: *Staphylococcus aureus* (ATCC 29213 and USA300LAC strains), *Streptococcus pneumoniae* (ATCC 49619 and 549 CHUAC strains), *Enterococcus faecalis* (ATCC 29212 and 256 CHUAC strains), and *Enterococcus faecium* (214 CHUAC strain).

Gram-negative and Gram-positive strains were routinely grown or maintained in Luria–Bertani (LB), and in Trypticase soya broth (TSB) media, respectively, supplemented with 2% agar or the antibiotic ampicillin (30 mg/L), when needed. All strains were grown at 37 °C and stored in 10% glycerol at −80 °C.

#### 3.6.2. Microdilution Method: Minimum Inhibitory Concentration

The minimum inhibitory concentrations (MICs) were determined by the broth microdilution method (Clinical Laboratory Standards Institute (CLSI), 2022). Briefly, the bacterial strains suspensions were cultured overnight at 37 °C in Mueller Hinton II agar plates (Becton Dickinson) and the turbidity of the bacterial suspensions was standarized at 0.5 on the McFarland scale to establish the inocula. The crude extracts of the test samples were dissolved in methanol. Two-fold serial dilutions of the extracts in Mueller Hinton II broth medium (Sigma) were carried out in 96-wells microplates, to produce a range of extract concentrations of 0.5–256 mg/L. Methanol was present at maximum concentration of 0.25% *v*/*v* in the well containing the highest concentration of compound. One well in each row contained growth media and bacterial suspension and was used as a positive growth control. Another well, containing medium only, was used as the negative control. Solvent controls of methanol and growth medium were included to determine whether the concentrations used interfered with bacterial growth. The MIC was evaluated after incubation 20–24 h to 37 °C and established as the lowest concentration of the compound at which the bacterial strains did not grow. All compounds were tested in triplicate.

As controls, MIC assays were also performed with imipenem against Gram-negative strains, and with vancomycin against Gram-positive strains (MIC values known).

## 4. Conclusions

In summary, three new diterpene alkaloids, (+)-8-epiagelasine T (**1**), (+)-10-epiagelasine B (**2**), and (+)-12-hydroxy agelasidine C (**3**), together with three known compounds, (+)-*ent*-agelasine F (**4**), (+)-agelasine B (**5**), and (+)-agelasidine C (**6**), were isolated from sponge *Agelas citrina* collected on Cozumel Island in the Mexican Caribbean. This is the first time that (+)-*ent*-agelasine F (**4**) is reported as a natural product. All of them display antibacterial activity against Gram-positive bacteria, (+)-10-epiagelasine B (**2**) being the most active one.

## Figures and Tables

**Figure 1 marinedrugs-20-00298-f001:**
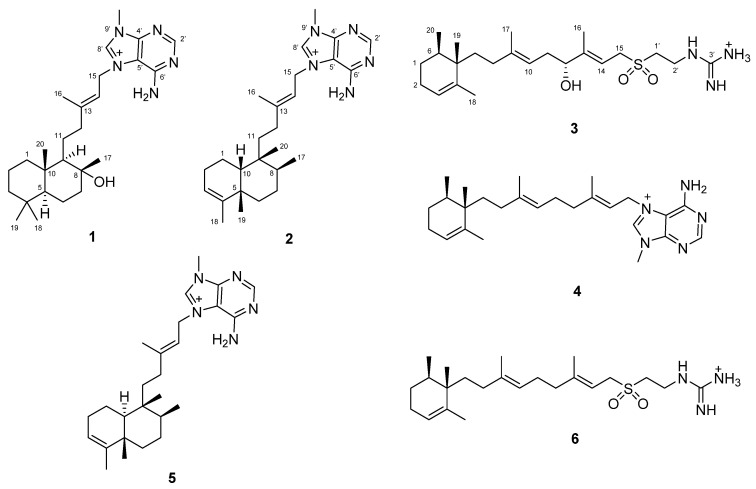
The structures of compounds **1**–**6** isolated from *Agelas citrina*.

**Figure 2 marinedrugs-20-00298-f002:**
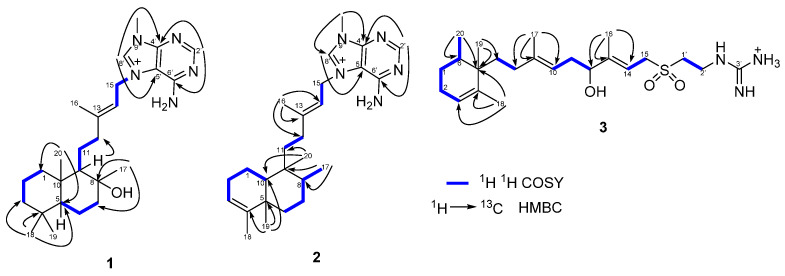
Key ^1^H-^1^H COSY and HMBC correlations of **1**–**3**.

**Figure 3 marinedrugs-20-00298-f003:**
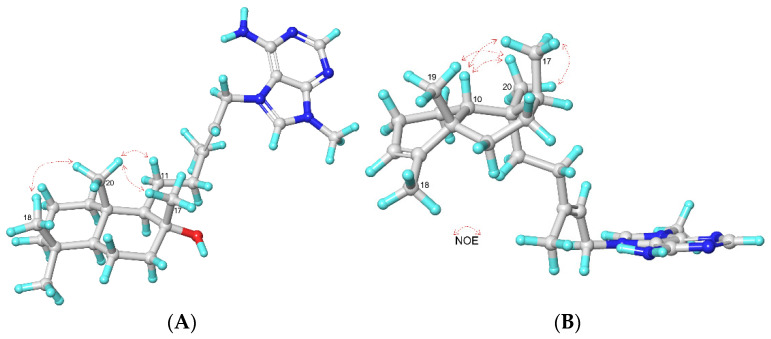
Key NOE correlations for compounds **1** (**A**) and **2** (**B**).

**Figure 4 marinedrugs-20-00298-f004:**
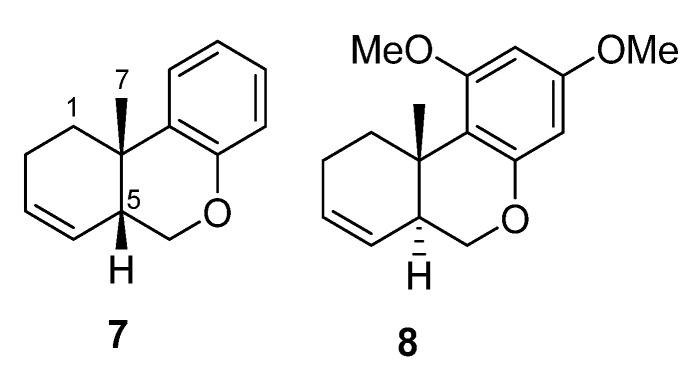
Structure of model compounds **7** and **8** used for the measurement of ^3^*J*_CH_ coupling constants.

**Figure 5 marinedrugs-20-00298-f005:**
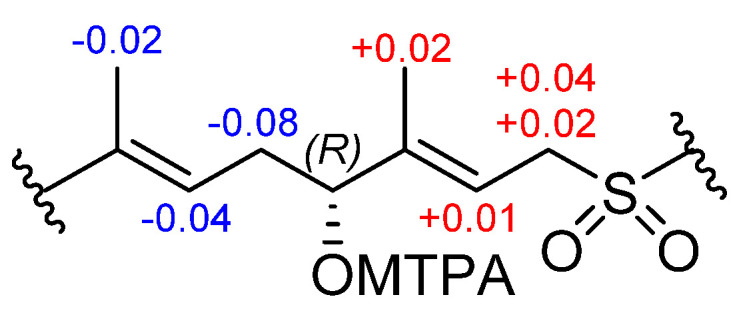
Δ*δ*^SR^ values (ppm) for the MTPA derivatives of **3** in CDCl_3_.

**Table 1 marinedrugs-20-00298-t001:** ^1^H NMR (500 MHz) and ^13^C NMR (125 MHz) data for 8-epiagelasine T (**1**), 10-epiagelasine B (**2**), and agelasine B (**5**).

Post.	1 *^a^*		2 *^b^*		5 *^b^*	
	*δ*_H_, Mult, (*J* in Hz)	*δ*_C_, Type	*δ*_H_, Mult, (*J* in Hz)	*δ*_C_, Type	*δ*_H_, Mult, (*J* in Hz)	*δ*_C_, Type
1	1.58, m 0.89, m	39.2, CH_2_	1.64 m 1.33, m	19.7, CH_2_	1.38, m 1.15, m	18.2, CH_2_
2	1.53, m	19.8, CH_2_	1.98, m	22.9, CH_2_	1.49, m	26.8, CH_2_
	1.19, m		1.91, m		1.29, m	
3	1.33, m	41.3, CH_2_	5.30, brs	120.7, CH	5.16, brs	120.5, CH
	1.10, m					
4		32.9, C		136.2, C		144.3, C
5	0.81, m	55.3, CH		40.5, C		38.3, C
6	1.34, m	17.7, CH_2_	1.80, m 1.71, m	33.7, CH_2_	1.70, d, (12.6) 1.14 m	36.7, CH_2_
7	1.70, dt, (11.8, 3.0) 1.30, m	43.7, CH_2_	1.64, m 1.42, m	29.2, CH_2_	1.43 m	27.4, CH_2_
8		72.1, C	1.31, m	33.5, CH	1.41, m	36.3, CH
9	0.98, m	59.9, CH		39.1, C		38.9, C
10		38.5, C	2.28, brd, (12.9)	42.0, CH	1.29 m	46.4, CH
11	1.55, m	22.7, CH_2_	1.35, m	30.0, CH_2_	1.60 m	32.5, CH_2_
	1.27, m		1.25, m		1.45 m	
12	2.22, td, (12.2, 6.1)	42.2, CH_2_	2.03, dt, (11.8, 5.6)	35.1, CH_2_	2.00 m	33.0, CH_2_
	2.09, td, (12.2, 4.9)		1.97, m		1.92 m	
13		147.0, C		149.2, C		148.8, C
14	5.44, t, (7.0)	114.6, CH	5.45, t, (6.4)	115.0, CH	5.43, t, (5.6)	115.0, CH
15	5.13, d, (7.3)	46.6, CH_2_	5.25, d, (6.5)	48.6, CH_2_	5.34, d, (6.0)	48.9, CH_2_
16	1.79, s	16.5, CH_3_	1.86, s	17.2, CH_3_	1.85, s	17.2, CH_3_
17	1.00, s	23.6, CH_3_	1.07, d, (7.4)	18.7, CH_3_	0.78, d, (5.9)	15.9, CH_3_
18	0.84, s	33.0, CH_3_	1.62, s	21.7, CH_3_	1.56, s	18.0, CH_3_
19	0.75, s	21.1, CH_3_	0.86, s	17.2, CH_3_	0.98, s	20.0, CH_3_
20	0.74, s	15.0, CH_3_	0.94, s	15.6, CH_3_	0.70, s	18.3, CH_3_
2′	8.46, s	155.4, CH	8.54, s	155.8, CH	8.52, s	155.2, CH
4′		149.0, C		149.7, C		149.6, C
5′		109.2, C		110.0, C		110.5, C
6′		152.4, C		151.8, C		151.7, C
8′	9.51, s	140.8, CH	9.87, s	142.2, C	10.06, s	142.3, CH
9′-*N*-Me	3.88, s	31.1, CH_3_	4.06, s	31.9, CH_3_	4.07, s	32.2, CH_3_
NH_2_	7.92, s					

*^a^* In DMSO-*d*_6_. *^b^* In CDCl_3_.

**Table 2 marinedrugs-20-00298-t002:** ^1^H NMR (500 MHz) and ^13^C (125 MHz) NMR data for 12-hydroxyagelasidine C (**3**) in CDCl_3_.

Position	3	
	*δ*_H_, Mult (*J* in Hz)	*δ*_C_, Type
1	1.44, m	27.2, CH_2_
2	1.95, m	25.7, CH_2_
3	5.40, brs	124.4, CH
4		139.7, C
5		40.5, C
6	1.71, m	33.3, CH
7	1.43, m	35.3, CH_2_
8	1.95, m; 1.48, m	34.6, CH_2_
9		140.1, C
10	5.06, t, (6.5)	118.6, CH
11	2.27, t, (6.5)	34.0, CH_2_
12	4.09, t, (6.5)	76.3, CH
13		149.2, C
14	5.52, t, (8.0)	110.0, CH
15	3.84, dd, (14.5, 8.0); 3.80, dd, (14.5, 8.0)	53.8, CH_2_
16	1.71, brs	13.1, CH_3_
17	1.61, brs	16.6, CH_3_
18	1.60, brs	19.3, CH_3_
19	0.84, s	16.0, CH_3_
20	0.85, d, (6.7)	21.2, CH_3_
1′	3.28, brs	50.3, CH_2_
2′	3.70, brs	34.5, CH_2_
3′		157.5, C

**Table 3 marinedrugs-20-00298-t003:** Antibacterial activity (MIC, μg/mL) of **1**–**6** against Gram-positive pathogens.

Compound	*S. aureus* ATCC 29213	*S. aureus* USA300LAC	*S. pneumoniae* ATCC 49619	*S. pneumoniae* 549 CHUAC	*E. faecalis* ATCC 29212	*E. faecalis* 256 CHUAC	*E. faecium* 214 CHUAC
(+)-8-epiagelasine T (**1**)	16	16	16	32	32	≥64	32
(+)-10-epiagelasine B (**2**)	1	2	4	8	4	4	4
(+)-12-hydroxyagelasidine C (**3**)	8	8	16	-	16	32	8
(+)-*ent*-agelasine F (**4**)	4	4	4	-	8	8	8
(+)-agelasine B (**5**)	2	2	4	16	8	8	4
(+)-agelasidine C (**6**)	8	8	4	-	8	8	8

## Data Availability

All datasets related to this article can be obtained from the authors.

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
