# Peer review of "Antimicrobial Diterpene Alkaloids from an Agelas citrina Sponge Collected in the Yucatán Peninsula"

_marinedrugs, 2022, doi:10.3390/md20050298_

Round 1

Reviewer 1 Report

Manuscript is clearly written and the detailed data were provided (as a Supplementary material) to support the findings. However, I have some suggestions/questions for Authors before acceptance for publication.

  • Abstract: line 20 „This represents the first time that 4 was isolated as natural product”  Meanwhile,  according to first sentence: „along with three known natural products…” – it seems that compound no 4 was known as a natural product. The same comment for Conclusion.
  • Line 22: „several bacterial pathogens” – specify wha t pathogens were tested. Moreover, it should be mentioned that compounds were not active against Gram-negative pathogens (with exception of agelasine B)
  • Introduction should give better background for investigation. Lines 41-45 summarized the study and it is unnecessary in Introduction.
  • Scheme for isolation included as a Supplementary material will facility to follow the procedure
  • Table 1: why in table  was described the characteristic of compound no 5 instead of compound 3? 3 was new isolated compound and compound 5 was previously isolated.

On the other hand, in line 186 is stated: “spectral data of 3 in CDCl3 (Table 1)”….

  • Line 266: “24 h each extraction. “ – maceration?
  • Line 293, 298, 305: “100% of CH3OH in H2O” – “in H2O“ is unnecessary
  • 6.1. Lack of italic for pathogens
  • Conclusion: add the name of compounds

Author Response

  1. Manuscript is clearly written, and the detailed data were provided (as a Supplementary material) to support the findings. However, I have some suggestions/questions for Authors before acceptance for publication.

Abstract: line 20 “This represents the first time that 4 was isolated as natural product”  Meanwhile, according to first sentence: „along with three known natural products…” – it seems that compound no 4 was known as a natural product. The same comment for Conclusion.

Answer to reviewer 1: The reviewer is right (4 is a known compound that was synthetized previously but this is the first time that it was isolated as a natural product). This was now clarified in the introduction and conclusions that.

  1. Line 22: “several bacterial pathogens” – specify what pathogens were tested. Moreover, it should be mentioned that compounds were not active against Gram-negative pathogens (with exception of agelasine B)

Answer to reviewer 1: We specified in the abstract what pathogens were tested, following the reviewer’s comment.

  1. Introduction should give better background for investigation. Lines 41-45 summarized the study and it is unnecessary in Introduction.

Answer to reviewer 1: Following the reviewer’s comment, the introduction was modified and a better background for the investigation was added.

  1. Scheme for isolation included as a Supplementary material will facility to follow the procedure

Answer to reviewer 1: Following the reviewer’s comment, a scheme for isolation is included in the supplementary material as scheme S1.

  1. Table 1: why in table was described the characteristic of compound no 5 instead of compound 3? 3 was new isolated compound and compound 5 was previously isolated.

Answer to reviewer 1: The NMR data in CDCl3 of 5 were included in Table 1 to visualize the comparison of these data to those of 2. Although the NMR data in CDCl3 of 5 are reported and the complete NMR assignment of 5 was also reported in other deuterated solvents, this is the first time that the complete assignment is reported in CDCl3. The NMR spectra of 3 is described in Table 2. 

  1. On the other hand, in line 186 is stated: “spectral data of 3 in CDCl3 (Table 1)”….

Answer to reviewer 1: This was a typographic error. This is Table 2 instead of Table 1 and it was corrected.

  1. Line 266: “24 h each extraction. “ – maceration?

Answer to reviewer 1: Following the reviewer’s comment, the kind of extraction used was specified as maceration.

  1. Line 293, 298, 305: “100% of CH3OH in H2O” – “in H2O“ is unnecessary

Answer to reviewer 1: Thank you for your recommendation and “in H2O” was deleted.

  1. 9. 6.1. Lack of italic for pathogens

Answer to reviewer 1: Following the reviewer’s comment, the name of pathogens were written in Italic.

  1. Conclusion: add the name of compounds

Answer to reviewer 1: The name of compounds is now included, following the reviewer’s comment.

Reviewer 2 Report

The Author should also consider also the following articles,  recently published in Natural Product Research.

1. Metabolomics approach to chemical diversity of the Mediterranean marine sponge Agelas oroides Pierre Sauleau, Céline Moriou & Ali Al Mourabit Natural Product Research, Volume 31, 2017 - Issue 14   2. Two New Naturally Occurring Pyrrole Derivatives from the Tropical Marine Sponge Agelas oroides Gabriele M. König & Anthony D. Wright Natural Product Letters, Volume 5, 1994 - Issue 2

Author Response

Comments to the Author

  1. The Author should also consider also the following articles, recently published in Natural Product Research.
  2. Metabolomics approach to chemical diversity of the Mediterranean marine sponge Agelas oroides Pierre Sauleau, Céline Moriou & Ali Al Mourabit Natural Product Research, Volume 31, 2017 - Issue 14
  3. Two New Naturally Occurring Pyrrole Derivatives from the Tropical Marine Sponge Agelas oroides Gabriele M. König & Anthony D. Wright Natural Product Letters, Volume 5, 1994 - Issue 2

Answer to reviewer 2: Thank you for your suggestion, the references were reviewed and inserted in the text.